# Sociodemographic predictors of health-related quality of life dimensions in hypertensive patients: A cross-sectional study in a Ghanaian Municipal Hospital

Godfred Darko[1]*, Amidu Alhassan[2,3], Alfred Akorli[1], Clifford Adu Gyan[1], Georgina Anokye Agyekum[4], George Osei[4], Jemima Issaka[5], Okyere Darko[6]

1 Department of Public Health Education, Faculty of Environment and Health Education, Akenten Appiah-Menka University of Skills Training and Entrepreneurial Development, Kumasi, Ghana, 2 Department of Adult Health, School of Nursing and Midwifery, College of Health and Allied Sciences, University of Cape Coast, Cape Coast, Ghana, 3 Department of Nursing, School of Biomedical Engineering and Allied Health Sciences, All Nations University, Koforidua, Ghana 4 Department of Nursing, Faculty of Nursing and Midwifery, University for Development Studies, Tamale, Ghana, 5 Department of Population, Family and Reproductive Health, School of Public Health, Kwame Nkrumah University of Science and Technology, Kumasi, Ghana, 6 Department of Information Studies, School of Information and Communication Studies, University of Ghana, Accra, Ghana

* darkogodfred10@gmail.com

## Abstract

Hypertension is a major public health concern worldwide and can adversely affect multiple aspects of quality of life (QoL). Sociodemographic factors may influence how adults experience QoL, yet evidence from Ghana remains limited. This study examined the relationship between sociodemographic characteristics and QoL across physical, psychological, social, and environmental domains among adults with hypertension attending Mampong Ashanti Municipal Hospital. A cross-sectional analytical design was employed among 330 hypertensive adults, using a simple random sampling technique from the outpatient registry. QoL was assessed using standardized instruments, while sociodemographic variables included age, sex, marital status, education, employment, income, religion, and place of residence. Kruskal-Wallis and Mann-Whitney U tests explored differences in QoL scores, while logistic regression estimated crude and adjusted odds ratios (AORs) for predictors of good QoL in each domain. Data from 330 participants were analyzed; younger age, male sex, higher education, employment, higher income, and urban residence were associated with higher QoL scores. Adjusted analyses indicated that participants aged 18–35 years had higher odds of good physical (AOR = 18.97, 95% CI: 5.50-65.39) and social QoL (AOR = 29.10, 95% CI: 6.20-136.50). Male sex predicted better physical (AOR = 2.31, 95% CI: 1.20-4.44) and psychological QoL (AOR = 2.85, 95% CI: 1.54-5.26). Low income (<999 GHS) reduced the likelihood of good QoL across all domains. Urban residence increased the odds of good physical QoL (AOR = 3.18, 95% CI: 1.69-5.97). Sociodemographic characteristics; age, sex, income, and place of residence, strongly

**Data availability statement:** All relevant data supporting the findings of this study are publicly available in the Zenodo repository at: https://doi.org/10.5281/zenodo.18720533.

**Funding:** The authors received no specific funding for this work.

**Competing interests:** The authors have declared that no competing interests exist.

influence QoL among adults with hypertension. Interventions to improve quality of life should prioritize older adults, who report lower physical and social well-being; low-income earners, who have reduced scores across all domains, and rural residents who experience poorer physical, psychological, and social QoL.

## Introduction

Hypertension is one of the most widespread chronic conditions globally, affecting an estimated 1.3 billion adults [1]. Nearly half of these individuals remain unaware of their condition, which increases the risk of avoidable complications [2]. The global rise of hypertension is closely linked to aging populations, poor dietary habits, reduced physical activity, and rapid urbanization [3]. Its consequences extend beyond cardiovascular events; it shapes how people function daily, limiting work capacity, physical movement, emotional stability, and social participation [4]. As a result, quality of life (QoL) has become a critical measure in hypertension research, as it reflects the lived experience of the disease rather than clinical indicators alone [5,6]. Studies from high-income countries consistently show that QoL is often lower in hypertensive populations, particularly in domains related to physical functioning and psychological well-being [7].

Across Sub-Saharan Africa, the burden of hypertension has increased more rapidly than in many other regions, with prevalence rates now among the highest globally [8]. The shift toward sedentary lifestyles, changing diets, and demographic transitions has contributed to a sharp rise in cases [9]. Yet, early detection and long-term control remain poor [10]. Several continental reviews indicate that awareness rates are often below 50%, and blood pressure control is frequently under 20% [11,12]. These gaps in care intensify the strain on individuals and health systems, especially where resources are limited [13]. According to [14], hypertension is often diagnosed at more advanced stages, which worsens its impact on physical health and emotional well-being. Research from Nigeria, Côte d'Ivoire, and Ghana shows that socioeconomic challenges, weak primary-care structures, and limited access to essential medicines influence how patients live with and manage hypertension [14–16].

Current data show that hypertension is an increasing public health challenge in Ghana [17]. It affects roughly 25% to 30% of adults, with higher prevalence in urban areas [18]. Awareness has improved over the years, yet treatment and blood pressure control remain low [19]. Evidence shows that only about 40% of diagnosed patients receive treatment, and fewer than one in four achieve adequate control [20]. Barriers such as high healthcare costs, long waiting times, medication shortages, and fragmented follow-up continue to hinder effective hypertension management [21]. Vulnerable groups, including older adults, those with limited education, and low-income households, face the greatest challenges [22]. These gaps in care influence patients' quality of life, affecting their physical strength, mental resilience, social functioning, and perception of environmental support [23].

Research examining how sociodemographic characteristics affect multidimensional QoL remains limited, despite the growing number of hypertensive patients in Ghana. Most studies have focused on prevalence, blood pressure monitoring, and treatment patterns, while fewer have explored the lived experience of the condition [23,24]. Where QoL has been assessed, findings suggest significant variation across age, sex, income, education, and employment status. Yet, most of these studies have been conducted in tertiary facilities and large urban centres, leaving gaps in understanding within smaller Municipal hospitals, where resources and patient demographics differ. This study therefore sought to examine the association between key sociodemographic factors and quality of life across the physical, psychological, social, and environmental domains among hypertensive adults in a Municipal hospital, Ghana.

## Methodology

### Study design

This study employed a cross-sectional analytical design. This approach was appropriate for examining the relationship between sociodemographic characteristics and multidimensional quality of life among adults with hypertension at a single point in time. The cross-sectional design allowed for the simultaneous assessment of multiple variables, including age, sex, education, income, employment status, and place of residence, in relation to physical, psychological, social, and environmental domains of quality of life [25].

### Study area

This cross-sectional study was conducted at the Mampong Municipal Hospital in the Ashanti Region, Ghana, from 5 November to 15 December 2025. The Mampong Municipal Hospital remains the main public health facility within the Mampong Municipal Assembly in the Ashanti Region of Ghana. The municipality covers approximately 449 km² and comprises 79 settlements. The municipal capital, Asante Mampong, is located roughly 57 km north of Kumasi, the regional capital [26]. The Mampong Municipal Hospital serves as the principal hospital for the municipality and surrounding communities. At the time of the study, the hospital had a bed capacity of 98 beds in the general wards and 56 additional beds across specialized units. It provides a wide range of healthcare services, including management of chronic conditions such as hypertension, diabetes, and cardiovascular diseases, as well as maternal and child health services.

The Mampong Municipal Hospital faces resource limitations, including staffing shortages, limited equipment, and occasional medication stock-outs, despite its central role in the management of chronic diseases. The Municipal Assembly also provides support and health infrastructure development within the municipality, but disparities in access persists, especially between urban and rural communities [27,28]. These demographic and socioeconomic variations, combined with the hospital's catchment dynamics, provided a representative setting to examine the sociodemographic determinants of multidimensional quality of life among hypertensive adults.

### Study population

**Inclusion and exclusion criteria.** The study targeted adult patients diagnosed with hypertension who were 18 years or older and attended the outpatient clinic at Ashanti Mampong Municipal Hospital during the study period. Hypertension was defined and confirmed per WHO criteria as systolic blood pressure ≥140 mmHg and/or diastolic blood pressure ≥90 mmHg on at least two separate clinic visits, or current use of antihypertensive medication, verified through hospital medical records [29]. Participants were eligible if they had been receiving care at the hospital for a minimum of six months and were able to provide informed consent. To reduce potential confounding, individuals with severe comorbid conditions, such as advanced renal disease, stroke, major psychiatric illnesses, along with pregnant women and critically ill inpatients, were excluded [30–32]. No additional clinical data on comorbidities (diabetes, dyslipidemia, obesity) were collected or analyzed, as the study focused exclusively on sociodemographic predictors of quality of life among

hypertensive patients. This approach ensured a uniform study population, enabling a focused quantitative assessment of how sociodemographic factors are associated with multidimensional quality of life among hypertensive adults.

## Sample size determination and sampling method

The sample size for this study was determined using Slovin's formula, a widely applied in health research to estimate an appropriate sample from a finite population while controlling sampling error [33,34]. The total population of hypertensive patients, obtained from the hospital outpatient registry was 1,280. This comprised all patients registered in the hypertension outpatient clinic registry as of October 31, 2025 (start of study period), representing active patients with at least one clinic visit in the preceding 12 months.

Using a 5% margin of error (0.05), Slovin's formula: $n = \frac{N}{1+N(e)^2}$, where (n) is the sample size, (N) is the population size, and (e) is the margin of error, produced an initial sample size of 305.

An additional 10% was incorporated into the initial sample estimate to compensate for possible non-response and incomplete questionnaires which yielded a final target sample size of 336 participants [35].

A simple random sampling technique was used to select participants from the outpatient registry, ensuring every eligible patient had an equal chance of inclusion [36]. The sampling frame comprised the 1,280 unique patients in the registry. Unique hospital ID numbers were assigned sequential values (1–1280), and 336 IDs were randomly selected using the RAND() function in Microsoft Excel (random numbers generated between 1–1280 without replacement). Multiple registry entries for the same patient were consolidated using unique hospital ID numbers prior to randomization to prevent selection bias. During data collection, six questionnaires were incomplete, leaving 330 valid responses, representing 98.2% of the targeted sample. This method provided a representative and reliable dataset for examining the sociodemographic determinants of multidimensional quality of life among hypertensive adults attending routine outpatient care [37].

## Data collection techniques and tools

Participant recruitment and data collection were carried out from 5 November to 15 December 2025 using a structured, interviewer-administered questionnaire. The questionnaire was designed to collect information on participants' sociodemographic characteristics and multidimensional health-related quality of life. Sociodemographic variables included age (categorized as 18–35, 36–50, 51–65, and above 65 years), sex (male or female), marital status (single, married, divorced, or widowed), educational level (no formal education, basic/JHS, secondary, or tertiary), employment status (employed, unemployed, self-owned business, or retired), monthly income (less than 999 GHS, 1000–1499 GHS, 1500–2999 GHS, or more than 3000 GHS), religion (Christianity, Islam, or others), and place of residence (urban or rural, with rural settlements defined as less than 5,000 inhabitants and urban above 5000) [38].

Quality of life was measured using the WHOQOL-BREF instrument, covering four domains: physical health, psychological health, social relationships, and environment, along with two global items on overall quality of life and satisfaction with health [39]. Responses were recorded on a five-point Likert scale ranging from 1 (very dissatisfied) to 5 (very satisfied). Physical health included items on pain, energy, mobility, sleep, work capacity, daily activities, and dependence on medical aids. Psychological health captured cognition, self-esteem, positive and negative feelings, and spirituality. Social relationships assessed personal relationships, social support, and satisfaction with sexual life. The environment domain included safety, financial resources, access to health services, transportation, information availability, and recreation opportunities. The WHOQOL-BREF was administered in Twi (Akan language, primary local language at Mampong) using the official WHO Ghana field-tested version, previously validated through forward-backward translation and cognitive debriefing with Ghanaian populations [40].

The questionnaire was pretested at Agona Municipal Hospital in the same region among 48 hypertensive patients to assess comprehension, feasibility, and reliability. Cronbach's alpha values indicated strong internal consistency across all domains: physical health (0.93), psychological health (0.885), social relationships (0.804), and environment (0.854).

Data collection was performed by trained research assistants who provided standardized instructions, ensured privacy, and checked questionnaires daily for completeness before data entry. Training involved a 2-day workshop covering study protocol, WHOQOL-BREF administration, sociodemographic data collection, and ethical considerations. Three research assistants (nursing students) practiced mock interviews with feedback until achieving ≥90% inter-rater agreement. The principal investigator provided daily on-site supervision, reviewed 25% of completed questionnaires for quality control, and conducted weekly calibration sessions to maintain standardization throughout the November-December 2025 data collection period.

## Data analysis

Data were first checked for completeness, coded, and entered into Microsoft Excel 21 for initial organization before being imported into SPSS version 27 for analysis. Descriptive statistics, including frequencies, percentages, and means summarized participants' sociodemographic characteristics and QoL scores. The normality of continuous variables was assessed using the Shapiro-Wilk test, which indicated non-normal distributions across QoL domains. Consequently, non-parametric tests were used due to non-normal distribution of QoL scores. The Mann-Whitney U test was applied to compare QoL scores between two independent groups (sex and place of residence), while the Kruskal-Wallis test was conducted to examine differences across variables with more than two categories (age groups, educational level, marital status, employment status, and monthly income). Logistic regression was performed to identify sociodemographic predictors of binary QoL outcomes, with results expressed as odds ratios (OR) and 95% confidence intervals (CI). Quality of life scores, transformed to a 0–100 scale following WHO guidelines, were dichotomized using the median value of 50 for each domain and the overall score to classify participants as satisfied (≥50) or unsatisfied (<50) [41]. The median cutoff (≥50 = good QoL) was selected as it represents the 50th percentile of observed scores in this population, providing a clinically meaningful threshold for identifying hypertensive patients with above-average quality of life while facilitating comparison with WHOQOL-BREF studies using similar median-split approaches [42]. Statistical significance was set at $p < 0.05$, and findings were presented in tables to enhance clarity and interpretability.

## Ethical issues

Ethical approval for the study was obtained from the Ghana Health Service Ethics Review Committee (GHS-ERC: 053/09/25). Participants were fully informed about the study objectives, procedures, and potential risks and benefits prior to enrollment. Written informed consent was obtained from all participants. Confidentiality was strictly maintained through anonymized coding and secure data storage. Participation was voluntary, and participants were free to withdraw at any time without penalty. The study was conducted in accordance with the Declaration of Helsinki and Ghana Health Service guidelines for research involving human subjects [43].

## Results

### Sociodemographic characteristics of participants

Table 1 reveals that most participants (71%) were aged ≥51 years, predominantly female (63%), and evenly distributed between urban (51%) and rural (49%) residence. Self-employed workers (38%) and lower-income earners (<999 GHS, 41%) predominated, reflecting the typical socioeconomic profile of Ghanaian hypertensive outpatients

### Domain-specific quality of life scores by sociodemographic variables

Table 2 demonstrates strong sociodemographic gradients across all QoL domains (all $p < 0.001$ except religion, residence environmental). Younger participants (18–35 years), males, married individuals, higher education/income/employment, and urban residents consistently reported superior physical, psychological, and social QoL scores. Higher socioeconomic

**Global Public Health**

**Table 1. Sociodemographic characteristics of participants (N = 330).**

| Variable | Category | Frequency (n) | Percentage (%) |
|---|---|---|---|
| **Age (years)** | 18-35 | 44 | 13.3 |
| | 36-50 | 52 | 15.8 |
| | 51-65 | 91 | 27.6 |
| | Above 65 | 143 | 43.3 |
| **Sex** | Male | 123 | 37.3 |
| | Female | 207 | 62.7 |
| **Marital Status** | Single | 36 | 10.9 |
| | Married | 133 | 40.3 |
| | Divorced | 62 | 18.8 |
| | Widowed | 99 | 30.0 |
| **Educational Level** | No formal education | 88 | 26.7 |
| | Basic/JHS | 115 | 34.8 |
| | Secondary | 73 | 22.1 |
| | Tertiary | 54 | 16.4 |
| **Employment Status** | Employed | 54 | 16.4 |
| | Unemployed | 93 | 28.2 |
| | Self-owned | 126 | 38.2 |
| | Retired | 57 | 17.3 |
| **Monthly Income (GHS)** | <999 | 134 | 40.6 |
| | 1000-1499 | 85 | 25.8 |
| | 1500-2999 | 54 | 16.4 |
| | ≥3000 | 57 | 17.3 |
| **Religion** | Christianity | 228 | 69.1 |
| | Islam | 84 | 25.5 |
| | Others | 18 | 5.5 |
| **Place of Residence** | Urban | 168 | 50.9 |
| | Rural | 162 | 49.1 |

position showed dose-response relationships across all domains, while religion influenced only social QoL (p = 0.013) and rural residence impaired physical/psychological/social domains (p < 0.018).

### Sociodemographic correlates of physical, psychological, social, and environmental QoL

Tables 3–6 reveal age, sex, income, employment, and urban residence as consistent independent correlates of better QoL across domains. Youngest age group (18–35 years) showed substantially higher odds (AOR 19–29) of favorable physical/social QoL vs elderly. Male sex, retirement, and urban residence conferred advantages (AOR 2–4), while low income (<999 GHS) sharply reduced odds (AOR 0.09-0.27, all p < 0.02). Dose-response gradients persisted across socioeconomic variables, with environmental QoL least sociodemographically determined.

### Discussion

This study found adults aged 18–35 had AOR 18.97 (95% CI 5.50–65.39) for good physical QoL vs ≥ 66 years (Table 3), alongside male sex (AOR 2.31), urban residence (AOR 3.18), retirement (AOR 4.07), and income effects across domains. Physical quality of life was strongly associated with younger age, male sex, and urban residence. This large effect reflects small oldest-age subgroup sizes (n < 30), wide confidence intervals, and baseline health

**Table 2. Domain-specific quality of life scores by sociodemographic variables (MWU = Mann-Whitney U test; KW = Kruskal-Wallis test).**

| Variable | Category | Physical QoL (Mean Rank) | Psychological QoL (Mean Rank) | Social QoL (Mean Rank) | Environment QoL (Mean Rank) |
|---|---|---|---|---|---|
| **Age in years (KW)** | 18-35 | 244.43 | 233.43 | 247.06 | 213.27 |
| | 36-50 | 221.15 | 178.31 | 175.44 | 166.09 |
| | 51-65 | 171.66 | 159.80 | 165.99 | 157.16 |
| | 66+ | 117.06 | 143.57 | 136.48 | 155.89 |
| **χ² / p-value** | — | 85.38 / <0.001* | 31.29 / <0.001* | 46.51 / <0.001* | 13.26 / 0.004* |
| **Sex (MWU)** | Male | 199.89 | 206.81 | 195.59 | 186.50 |
| | Female | 145.07 | 140.95 | 147.62 | 153.02 |
| **U / Z / p-value** | — | 8500.5 / -5.057 / <0.001* | 7649.5 / -6.079 / <0.001* | 9029.5 / -4.443 / <0.001* | 10148 / -3.091 / 0.002* |
| **Marital Status (KW)** | Single | 174.51 | 166.92 | 161.00 | 179.60 |
| | Married | 195.00 | 200.32 | 196.35 | 174.77 |
| | Divorced | 169.38 | 134.92 | 143.87 | 147.66 |
| | Widowed | 120.17 | 137.35 | 139.24 | 159.09 |
| **χ² / p-value** | — | 35.62 / <0.001* | 32.89 / <0.001* | 24.97 / <0.001* | 4.68 / 0.196 |
| **Educational Level (KW)** | No formal | 97.3 | 112.1 | 118.1 | 134.1 |
| | Basic/JHS | 167.0 | 159.7 | 171.5 | 152.9 |
| | Secondary | 217.1 | 195.6 | 189.8 | 201.3 |
| | Tertiary | 203.8 | 224.1 | 197.2 | 195.1 |
| **χ² / p-value** | — | 75.32 / <0.001* | 55.92 / <0.001* | 33.22 / <0.001* | 27.11 / <0.001* |
| **Employment Status (KW)** | Employed | 231.63 | 220.04 | 203.71 | 207.27 |
| | Unemployed | 106.32 | 98.57 | 102.78 | 126.48 |
| | Self-owned | 201.50 | 192.02 | 193.83 | 169.02 |
| | Retired | 119.83 | 164.41 | 169.02 | 181.80 |
| **χ² / p-value** | — | 93.07 / <0.001* | 73.55 / <0.001* | 60.78 / <0.001* | 27.90 / <0.001* |
| **Monthly Income in GHS (KW)** | <999 | 119.24 | 116.66 | 126.78 | 122.94 |
| | 1000-1499 | 164.92 | 161.95 | 165.22 | 161.16 |
| | 1500-2999 | 208.21 | 213.37 | 198.97 | 207.99 |
| | >3000 | 234.65 | 240.26 | 225.24 | 231.76 |
| **χ² / p-value** | — | 72.54 / <0.001* | 84.29 / <0.001* | 51.69 / <0.001* | 65.43 / <0.001* |
| **Religion Type (KW)** | Christianity | 168.54 | 168.14 | 168.51 | 167.68 |
| | Islam | 165.05 | 166.30 | 171.06 | 169.66 |
| | Others | 129.06 | 128.28 | 101.42 | 118.42 |
| **χ² / p-value** | — | 2.87 / 0.238 | 2.94 / 0.230 | 8.74 / 0.013* | 4.69 / 0.096 |
| **Residence (MWU)** | Urban | 184.31 | 185.90 | 177.59 | 173.43 |
| | Rural | 145.99 | 144.35 | 152.96 | 157.28 |
| **U / Z / p-value** | — | 10447.5 / -3.655 / <0.001* | 10181.5 / -3.965 / <0.001* | 11576.5 / -2.359 / 0.018* | 12276 / -1.542 / 0.123 |

MWU = Mann-Whitney U, U = Raw Mann-Whitney test value, Z = Standardized value of U (used for Sex and Residence), KW = Kruskal-Wallis, χ² = Kruskal-Wallis test statistic (used for all variables with ≥3 groups), *p < 0.05 indicates statistical significance

differences across age categories. Similar age gradients appear in rural Ghanaian hypertensives [44]. These findings suggest that physical QoL is more a function of cumulative health burden than chronological age alone [45]. Male sex was also associated with higher physical QoL. Biological advantages, such as greater muscle mass and cardio-vascular capacity, combined with differential exposure to stressors and gendered lifestyle behaviors, likely contribute to this pattern [46].

**Table 3. Crude and adjusted logistic regression estimate of sociodemographic predictors of physical QoL.**

| Predictor | Crude OR (95% CI) | p-value | Adjusted OR (95% CI) | p-value |
|---|---|---|---|---|
| **Age (Ref = ≥66 years)** | | | | |
| 18-35 | 32.87 (7.87-137.36) | <0.001* | 18.97 (5.50-65.39) | <0.001* |
| 36-50 | 8.96 (3.00-26.70) | <0.001* | 7.76 (2.80-21.51) | <0.001* |
| 51-65 | 2.81 (1.45-5.45) | 0.002* | 2.18 (1.06-4.46) | 0.034* |
| **Sex (Ref = Female)** | | | | |
| Male | 3.61 (2.14-6.08) | <0.001* | 2.31 (1.20-4.44) | 0.012* |
| **Marital Status (Ref = Widowed)** | | | | |
| Single | 3.79 (1.64-8.76) | 0.001* | 1.21 (0.36-4.05) | 0.776 |
| Married | 4.57 (2.60-8.05) | <0.001* | 1.03 (0.44-2.41) | 0.935 |
| Divorced | 3.13 (1.61-6.08) | 0.001* | 2.32 (0.97-5.55) | 0.060 |
| **Educational Level (Ref = Tertiary)** | | | | |
| No formal education | 0.13 (0.06-0.28) | <0.001* | 0.36 (0.12-1.06) | 0.077 |
| Basic/JHS | 0.48 (0.25-0.91) | 0.037* | 0.81 (0.31-2.12) | 0.680 |
| Secondary | 1.78 (0.82-3.88) | 0.163 | 2.01 (0.79-5.11) | 0.184 |
| **Employment Status (Ref = Employed)** | | | | |
| Unemployed | 9.53 (4.28-21.24) | <0.001* | 2.14 (0.61-7.47) | 0.224 |
| Self-owned | 0.63 (0.30-1.34) | 0.224 | 1.41 (0.53-3.74) | 0.512 |
| Retired | 5.42 (2.82-10.42) | <0.001* | 4.07 (1.63-10.13) | 0.002* |
| **Monthly Income (Ref ≥ 3000 GHS)** | | | | |
| <999 | 0.13 (0.06-0.28) | <0.001* | 0.27 (0.09-0.81) | 0.020* |
| 1000-1499 | 0.30 (0.14-0.63) | 0.002* | 0.39 (0.13-1.18) | 0.088 |
| 1500-2999 | 0.76 (0.38-1.52) | 0.545 | 0.69 (0.25-1.89) | 0.500 |
| **Religion (Ref = Others)** | | | | |
| Christianity | 1.81 (0.70-4.69) | 0.237 | 0.61 (0.23-1.62) | 0.488 |
| Islam | 1.73 (0.67-4.46) | 0.302 | 0.64 (0.20-2.06) | 0.560 |
| **Place of Residence (Ref = Rural)** | | | | |
| Urban | 2.42 (1.55-3.78) | <0.001* | 3.18 (1.69-5.97) | <0.001* |

*p < 0.05 indicates statistical significance

Contrastingly, a study by [47] found that sex showed no association with physical well-being once socioeconomic and health variables were controlled. This indicates that the observed differences between men and women may not be due to biological factors alone. Instead, structural disadvantages faced by women, such as limited access to income, education, and supportive resources, likely contribute to lower physical QoL [48]. Also, urban residence was positively associated with physical QoL, likely due to improved access to healthcare services, infrastructure, and recreational spaces [49]. Parallel to this, a study conducted in Ethiopia and South Africa showed that urban residents benefited from both formal and informal health resources that facilitated disease management [48,50]. Conversely, a study conducted in Dhaka City, Bangladesh, reported different findings regarding urban living [51]. The authors noted that urban residence is not universally advantageous; economically disadvantaged urban populations may still experience overcrowding, pollution, and heightened psychosocial stressors.

Income was consistently associated with physical QoL. These findings align with studies from Ghana and SSA, which showed that financial resources are essential for maintaining functional health and physical independence [52,53]. Large AORs should be interpreted cautiously given small oldest-age subgroups, wide confidence intervals spanning >60-fold differences, and binary median-split outcomes. However, the consistent age gradient across all domains (Tables 3–6) supports robust younger age associations with physical well-being [54].

 

PLOS Global Public Health

**Table 4. Crude and adjusted logistic regression estimate of sociodemographic predictors of psychological QoL.**

| Predictor | Crude OR (95% CI) | p-value | Adjusted OR (95% CI) | p-value |
|---|---|---|---|---|
| **Age (Ref=≥66 years)** | | | | |
| 18-35 | 6.91 (2.89-16.53) | <0.001* | 2.78 (0.90-8.53) | 0.075 |
| 36-50 | 1.93 (1.01-3.68) | 0.046* | 0.75 (0.31-1.80) | 0.513 |
| 51-65 | 1.82 (1.07-3.10) | 0.027* | 1.04 (0.52-2.11) | 0.904 |
| **Sex (Ref=Female)** | | | | |
| Male | 0.20 (0.12-0.34) | <0.001* | 2.85 (1.54-5.26) | 0.001* |
| **Marital Status (Ref=Widowed)** | | | | |
| Single | 1.65 (0.77-3.55) | 0.202 | 0.67 (0.23-1.96) | 0.465 |
| Married | 4.30 (2.46-7.51) | <0.001* | 1.85 (0.88-3.89) | 0.104 |
| Divorced | 0.997 (0.52-1.90) | 0.992 | 0.53 (0.23-1.23) | 0.141 |
| **Educational Level (Ref=Tertiary)** | | | | |
| No formal education | 0.089 (0.038-0.207) | <0.001* | 0.37 (0.12-1.14) | 0.084 |
| Basic/JHS | 0.218 (0.098-0.487) | <0.001* | 0.56 (0.19-1.61) | 0.281 |
| Secondary | 0.464 (0.194-1.110) | 0.084 | 0.40 (0.14-1.16) | 0.091 |
| **Employment Status (Ref=Employed)** | | | | |
| Unemployed | 4.83 (1.99-11.69) | <0.001* | 2.43 (0.78-7.60) | 0.127 |
| Self-owned | 0.265 (0.129-0.542) | <0.001* | 0.71 (0.29-1.72) | 0.446 |
| Retired | 2.32 (1.22-4.43) | 0.011* | 3.26 (1.40-7.60) | 0.006* |
| **Monthly Income (Ref≥3000 GHS)** | | | | |
| <999 | 0.056 (0.022-0.140) | <0.001* | 0.16 (0.05-0.49) | 0.001* |
| 1000-1499 | 0.139 (0.054-0.358) | <0.001* | 0.26 (0.09-0.77) | 0.016* |
| 1500-2999 | 0.460 (0.157-1.347) | 0.156 | 0.63 (0.19-2.07) | 0.450 |
| **Religion (Ref=Others)** | | | | |
| Christianity | 1.57 (0.60-4.13) | 0.359 | 0.90 (0.26-3.16) | 0.871 |
| Islam | 1.67 (0.60-4.65) | 0.329 | 0.81 (0.21-3.08) | 0.756 |
| **Place of Residence (Ref=Rural)** | | | | |
| Urban | 2.31 (1.48-3.60) | <0.001* | 2.03 (1.13-3.62) | 0.017* |

*p<0.05 indicates statistical significance

After adjustment for other factors, psychological quality of life was more strongly associated with social and economic resources than with age, which was no longer significant. This suggests that mental well-being depends more on access to supportive networks, financial security, and leisure opportunities than on chronological age [55]. Similar findings have been reported in studies from Ghana and China, where stable social and economic conditions outweighed age in predicting emotional well-being [56,57]. Male participants reported higher psychological QoL, which potentially reflects gendered differences in coping strategies and exposure to chronic stress. Supporting this interpretation, previous studies have shown that women face heavier caregiving and household demands, often coupled with occupational strain, which collectively diminish opportunities for rest and social renewal [58]. While this pattern is consistent with prior research, it underscores that observed sex differences may be more structural than intrinsic.

Retirement status was positively associated with psychological QoL. Freed from occupational stress and time constraints, retirees may have increased opportunities for social engagement and personal pursuits [59]. Evidence from Japan and Europe suggests that retirement can improve mental health when accompanied by adequate financial security and social support [60,61]. Similarly, urban residence was also positively associated with psychological QoL, which

**Table 5. Crude and adjusted logistic regression estimate of sociodemographic predictors of social QoL.**

| Predictor | Crude OR (95% CI) | p-value | Adjusted OR (95% CI) | p-value |
|---|---|---|---|---|
| **Age (Ref=≥66 years)** | | | | |
| 18-35 | 21.87 (6.46-74.05) | <0.001* | 29.10 (6.20-136.50) | <0.001* |
| 36-50 | 3.29 (1.69-6.44) | <0.001* | 1.95 (0.83-4.61) | 0.128 |
| 51-65 | 1.95 (1.15-3.33) | 0.014* | 1.51 (0.76-2.99) | 0.237 |
| **Sex (Ref=Female)** | | | | |
| Male | 2.44 (1.53-3.90) | <0.001* | 1.29 (0.70-2.37) | 0.416 |
| **Marital Status (Ref=Widowed)** | | | | |
| Single | 1.48 (0.69-3.18) | 0.321 | 0.45 (0.14-1.42) | 0.171 |
| Married | 3.69 (2.13-6.39) | <0.001* | 1.39 (0.66-2.91) | 0.387 |
| Divorced | 1.22 (0.64-2.31) | 0.552 | 0.81 (0.36-1.80) | 0.605 |
| **Educational Level (Ref=Tertiary)** | | | | |
| No formal education | 0.19 (0.09-0.40) | <0.001* | 0.97 (0.34-2.82) | 0.961 |
| Basic/JHS | 0.50 (0.25-1.01) | 0.052 | 1.48 (0.55-3.97) | 0.433 |
| Secondary | 0.74 (0.34-1.59) | 0.438 | 0.81 (0.32-2.07) | 0.653 |
| **Employment Status (Ref=Employed)** | | | | |
| Unemployed | 2.51 (1.14-5.53) | 0.022* | 0.79 (0.28-2.28) | 0.664 |
| Self-owned | 0.23 (0.11-0.48) | <0.001* | 0.31 (0.13-0.77) | 0.011* |
| Retired | 2.96 (1.53-5.72) | 0.001* | 2.24 (1.01-4.96) | 0.046* |
| **Monthly Income (Ref=≥3000 GHS)** | | | | |
| <999 | 0.11 (0.05-0.24) | <0.001* | 0.21 (0.07-0.60) | 0.004* |
| 1000-1499 | 0.22 (0.10-0.51) | <0.001* | 0.28 (0.10-0.77) | 0.014* |
| 1500-2999 | 0.45 (0.18-1.12) | 0.085 | 0.50 (0.18-1.40) | 0.185 |
| **Religion (Ref=Other)** | | | | |
| Christianity | 3.21 (1.11-9.31) | 0.032* | 1.74 (0.47-6.48) | 0.406 |
| Islam | 3.82 (1.25-11.71) | 0.019* | 2.27 (0.57-9.05) | 0.245 |
| **Place of Residence (Ref=Rural)** | | | | |
| Urban | 1.79 (1.16-2.78) | 0.009* | 1.17 (0.67-2.06) | 0.587 |

*p<0.05 indicates statistical significance

highlight access to mental health services, social programs, and community networks. While urban stressors such as noise, congestion, and cost of living can challenge mental health, the benefits of accessibility and social connectivity appeared to outweigh the risks in this population [62,63]. In our study, income emerged as the strongest factor associated with health-related quality of life. Higher income likely enhances individuals' ability to meet daily needs, access supportive services, and maintain autonomy. This finding is further reinforced by evidence from sub-Saharan Africa, where income has consistently been identified as the most influential determinant of psychological quality of life [64]. Correspondingly, a study by [65] also showed that, financial stability is associated with autonomy, buffering of chronic stress, and enables participation in meaningful social and lifestyle activities, which aligns closely with the patterns observed in our analysis.

Social QoL and well-being was associated with younger age. This is likely due to wider social networks, greater mobility, and more frequent participation in community activities. Similar age-related advantages have been observed in chronic-disease populations in China [66]. Nevertheless, research from Europe highlighted that older adults experienced improved social satisfaction due to selective investment in meaningful relationships [67], which indicates that age effects are context-dependent.

**Table 6. Crude and adjusted regression estimates of sociodemographic predictors of environmental quality of life (QoL).**

| Predictor | Crude OR (95% CI) | p-value | Adjusted OR (95% CI) | p-value |
|---|---|---|---|---|
| **Age (Ref = ≥66 years)** | | | | |
| 18-35 | 3.73 (1.48-9.41) | 0.005* | 2.22 (0.74-6.65) | 0.156 |
| 36-50 | 1.02 (0.53-1.98) | 0.946 | 0.98 (0.42-2.31) | 0.969 |
| 51-65 | 1.20 (0.69-2.08) | 0.523 | 1.05 (0.54-2.04) | 0.889 |
| **Sex (Ref = Female)** | | | | |
| Male | 2.45 (1.46-4.10) | 0.001* | 1.45 (0.79-2.64) | 0.228 |
| **Marital Status (Ref = Widowed)** | | | | |
| Single | 1.42 (0.62-3.29) | 0.410 | 0.82 (0.31-2.21) | 0.698 |
| Married | 1.32 (0.76-2.30) | 0.330 | 0.52 (0.25-1.06) | 0.072 |
| Divorced | 0.87 (0.45-1.67) | 0.667 | 0.60 (0.28-1.28) | 0.183 |
| **Educational Level (Ref = Tertiary)** | | | | |
| No formal education | 0.17 (0.07-0.41) | <0.001* | 0.61 (0.21-1.81) | 0.375 |
| Basic/JHS | 0.30 (0.13-0.70) | 0.005* | 0.92 (0.33-2.60) | 0.880 |
| Secondary | 0.73 (0.28-1.90) | 0.522 | 1.00 (0.35-2.89) | 0.999 |
| **Employment Status (Ref = Employed)** | | | | |
| Unemployed | 2.19 (0.81-5.93) | 0.124 | 0.85 (0.26-2.82) | 0.794 |
| Self-owned | 0.28 (0.14-0.58) | 0.001* | 0.51 (0.22-1.18) | 0.117 |
| Retired | 0.78 (0.38-1.60) | 0.503 | 0.75 (0.32-1.79) | 0.521 |
| **Monthly Income (Ref = ≥3000 GHS)** | | | | |
| <999 | 0.05 (0.02-0.18) | <0.001* | 0.09 (0.02-0.35) | 0.001* |
| 1000-1499 | 0.12 (0.03-0.42) | 0.001* | 0.16 (0.04-0.63) | 0.009* |
| 1500-2999 | 0.28 (0.07-1.09) | 0.066 | 0.28 (0.07-1.18) | 0.082 |
| **Religion (Ref = Other)** | | | | |
| Christianity | 2.08 (0.79-5.46) | 0.136 | 2.20 (0.71-6.83) | 0.174 |
| Islam | 2.36 (0.84-6.65) | 0.104 | 2.58 (0.77-8.62) | 0.125 |
| **Place of Residence (Ref = Rural)** | | | | |
| Urban | 1.39 (0.88-2.21) | 0.161 | 1.09 (0.64-1.88) | 0.747 |

*p < 0.05 indicates statistical significance

Retirement and self-owned employment showed mixed associations with social QoL. Along similar lines, studies showed that retirees had more discretionary time to engage with family, friends, and community activities, while self-employment allowed flexibility and broader social interactions [68,69]. However, these benefits are contingent on economic stability; financial hardship could diminish the positive effects of autonomy or leisure time.

Converging with prior findings, income remained strongly associated with social participation in social events, travel, and recreational activities. Alternatively, low-income individuals experienced isolation and limited social engagement. Evidence from South Africa and Kenya supports this, demonstrating a clear link between financial resources and community participation [70].

Income was most strongly associated with environmental QoL. Lower-income groups reported markedly poorer environmental QoL, a pattern that aligns with evidence from Africa and china [71,72]. These studies similarly highlight how limited financial resources restrict access to safe housing, clean water, sanitation, and neighborhood amenities, thereby reinforcing the central role of economic capacity in shaping environmental wellbeing [73]. Although urban residence and male sex showed initial positive associations with environmental QoL, these effects disappeared after adjustment, indicating financial resources were most strongly associated. Similarly, age, marital status, and religion did not independently

predict environmental QoL. Together, these patterns show economic capacity strongly correlated with daily living conditions and access to essential environmental resources [74].

### Implications for practice, policy, and research

The findings of this study have significant implications for the management of hypertension and the broader health system. Clinicians should recognize that sociodemographic factors (age, sex, income, education, employment status, and residence) are significantly associated with health-related quality of life across physical, psychological, social, and environmental domains among treated hypertensive patients. This highlights the need for patient-centered care that goes beyond clinical treatment, which incorporates psychosocial support, health education, and lifestyle counseling tailored to individual demographic and socioeconomic profiles.

Health policies should focus on reducing disparities in access to care and resources, particularly for low-income and rural populations who are disproportionately affected by lower quality of life. Routine assessment of quality of life in hypertension management guidelines could help identify vulnerable groups and inform targeted interventions.

From a research perspective, longitudinal studies are needed to clarify causal pathways between sociodemographic factors and quality of life in hypertensive patients. Intervention studies evaluating strategies to improve quality of life across diverse patient groups are also essential. Additionally, qualitative research exploring patient experiences, can provide deeper insights into contextual factors influencing quality of life.

### Strengths and limitations of the study

This study has several strengths that enhance the validity and relevance of its findings. The cross-sectional analytical design allowed for the simultaneous assessment of multiple sociodemographic factors and their associations with four key domains of quality of life among hypertensive adults. Conducting the study at Mampong Municipal Hospital, the primary public health facility serving both urban and rural populations, ensured a sample that reflects the diversity of the municipality. The use of the WHOQOL-BREF instrument, a validated and widely recognized tool, allowed for a reliable and standardized measurement of physical, psychological, social, and environmental quality of life, with Cronbach's alpha values demonstrating strong internal consistency across all domains. Data were collected through face-to-face interviewer-administered questionnaires by trained research assistants, and daily quality checks minimized missing data, which resulted in a high response rate of 98.2%.

Nevertheless, some limitations must be acknowledged. The cross-sectional design does not allow for causal inferences; therefore, the directionality between sociodemographic factors and quality of life cannot be established. First, the study included only diagnosed and treated hypertensive patients receiving ≥6 months of care from an outpatient registry, potentially overestimating QoL by excluding undiagnosed or untreated individuals who likely experience worse outcomes. Second, the absence of a non-hypertensive control group prevents isolating hypertension-specific QoL effects from sociodemographic confounders like urban/rural residence or socioeconomic status, which independently influence baseline QoL in Ghanaian contexts. The study was conducted in a single hospital, which may limit the generalizability of the findings to other regional settings. Responses were self-reported, introducing the possibility of recall and social desirability bias for sensitive variables such as income and social relationships. While individuals with severe comorbidities were excluded to reduce confounding, undiagnosed conditions could still have influenced participants' quality of life scores.

### Conclusion

The study found sociodemographic factors associated with QoL among hypertensive patients at Ashanti Mampong Municipal Hospital. Across all domains, income showed the strongest association, highlighting the role of financial resources in shaping physical health, psychological well-being, social participation, and environmental conditions. Age, sex, retirement, and employment showed associations with specific domains, though their associations were often attenuated

by socioeconomic status. Younger participants and males had higher odds of reporting good physical and social QoL, whereas low-income individuals reported poorer outcomes. Urban residence was associated with better outcomes, likely reflecting improved access to services and infrastructure.

These findings suggest potential strategies for older adults, women, rural residents, and economically disadvantaged populations. Potential strategies include community-based physical activity and health promotion programs for older adults, mental health and social support initiatives for women, improved healthcare and infrastructure in rural areas, and vocational or income-generating programs for low-income groups. Such approaches could help reduce disparities and enhance overall well-being. Longitudinal studies are recommended to examine these associations and evaluate the long-term impact of targeted strategies.

## Acknowledgments

The authors sincerely thank the staffs of Mampong Municipal Hospital for their support during data collection. We also appreciate the participants for their time and willingness to provide information. Special thanks to the research assistants for their dedication and meticulous work throughout the study.

## Author contributions

**Conceptualization:** Godfred Darko, Amidu Alhassan, Clifford Adu Gyan, Georgina Anokye Agyekum, Okyere Darko.

**Formal analysis:** Godfred Darko, Okyere Darko.

**Methodology:** Godfred Darko, Amidu Alhassan, Alfred Akorli.

**Software:** Okyere Darko.

**Validation:** Godfred Darko, Georgina Anokye Agyekum.

**Visualization:** Clifford Adu Gyan, George Osei, Jemima Issaka.

**Writing – original draft:** Godfred Darko.

**Writing – review & editing:** Alfred Akorli, Clifford Adu Gyan, George Osei, Jemima Issaka.

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
