## [Decision Letter · Decision Letter 0]

19 Feb 2026

PGPH-D-25-04048

Sociodemographic predictors of health-related quality of life dimensions in hypertensive patients: A cross-sectional study in a Ghanaian Municipal Hospital

Dear Dr. Darko,

Thank you for submitting your manuscript to PLOS Global Public Health. After careful consideration, we feel that it has merit but does not fully meet PLOS Global Public Health’s publication criteria as it currently stands. Therefore, we invite you to submit a revised version of the manuscript that addresses the points raised during the review process.

We look forward to receiving your revised manuscript.

Kind regards,

Theingi Maung Maung, Ph.D

Academic Editor

Journal Requirements:

1. Please amend your online Financial Disclosure statement. If you did not receive any funding for this study, please simply state: "The authors received no specific funding for this work."

2. In the online submission form, you indicated that "The datasets underpinning this work are accessible upon reasonable request from the corresponding author.".

a) In a public repository,

b) Within the manuscript itself, or

d) Uploaded as supplementary information.

For further assistance, you may go to: http://journals.plos.org/globalpublichealth/s/data-availability

Additional Editor Comments (if provided):

Reviewers' comments:

Reviewer's Responses to Questions

**Comments to the Author**

1. Does this manuscript meet PLOS Global Public Health’s publication criteria?

Reviewer #1: Yes

Reviewer #2: Partly

2. Has the statistical analysis been performed appropriately and rigorously?

Reviewer #1: Yes

Reviewer #2: Yes

3. Have the authors made all data underlying the findings in their manuscript fully available (please refer to the Data Availability Statement at the start of the manuscript PDF file)?

Reviewer #1: Yes

Reviewer #2: Yes

4. Is the manuscript presented in an intelligible fashion and written in standard English?

Reviewer #1: Yes

Reviewer #2: Yes

Reviewer #1: Methodology

• Please clarify how hypertension was defined and confirmed in this study.

• Please clarify whether patients with common comorbidities (e.g., diabetes, dyslipidemia, obesity) were included, since these conditions may affect quality of life and act as confounding factors.

• Please explain how the total population (N = 1,280) was identified (e.g., all registered hypertensive OPD patients in 2025, active follow-up patients only, or those attending the clinic during the study period).

• Please describe the simple random sampling process in more detail, including the sampling frame, the random selection method, and how repeated visits or duplicate records were handled.

• Please clarify how interviewers were trained and supervised, and how data collection was standardized to ensure consistency.

• Please explain whether the WHOQOL-BREF was used in the original language or translated and describe the translation process and any cultural adaptation.

Results

• Please revise the Results section to highlight the main findings from each table, rather than repeating all numbers already shown in the tables.

Discussion

• Please avoid causal wording; since this is a cross-sectional study, consider using “associated factors” instead of “predictors.”

Reviewer #2: Major Comments: reform the limitations section

The introduction highlights that globally, nearly half of hypertensive individuals are unaware of their condition [1,2], and in Ghana, only about 40% of diagnosed patients receive treatment, with fewer than 25% achieving adequate control [19,20]. However, by recruiting participants exclusively from the outpatient registry and requiring them to have received care for at least six months, the study only captures the experiences of diagnosed and treated individuals. This misses the quality of life (QoL) perspectives of undiagnosed or inadequately treated populations, limiting the representativeness of the findings for the broader hypertensive population in Ghana. I recommend adding this to the limitations section and discussing how it might lead to an overestimation of QoL, since untreated individuals are likely to have worse outcomes.

The absence of a non-hypertensive control group makes it difficult to isolate the specific impact of hypertension on QoL from underlying sociodemographic influences, such as urban/rural residence acting as a confounder. As noted, populations in sub-Saharan Africa may already have lower baseline QoL due to socioeconomic factors, regardless of hypertension. The cross-sectional design, which is already mentioned in the limitations, further complicates this by preventing the separation of confounders or the establishment of causality. Please expand the limitations section to explicitly address this issue, and suggest that future studies incorporate matched control groups or longitudinal designs to better disentangle these effects. Additionally, in the discussion, consider softening language about hypertension's role by using terms like "associated with" instead of implying direct influence.

Minor Comments

There are inconsistencies in the income categories across the text and tables (e.g., "<999 GHS" is used in most sections, but "<500 GHS" appears in the environmental QoL results). Please standardize these for clarity.

Some phrases imply directionality or causality (e.g., "influence" in the conclusion), which doesn't align with the cross-sectional design's limitations. Revise these to use "association" instead.

Please verify the accuracy of citations dated 2025 (e.g., [7,10,44]), considering the study's 2025 period, and shorten any long URLs to DOIs where possible.

Address minor grammar and clarity issues, such as changing "aORs" to "AORs" for consistency, correcting "physical QoL is more, a function" to "physical QoL is more a function," and adding citations for any uncited claims in the discussion.

In the methods section, briefly explain the rationale for using the median cutoff (≥50 = good QoL) rather than other thresholds.

In the table captions, define all abbreviations (e.g., KW for Kruskal-Wallis) to improve readability.

**Do you want your identity to be public for this peer review?** For information about this choice, including consent withdrawal, please see our Privacy Policy

Reviewer #1: No

Reviewer #2: **Yes:** Dr Syed Irfan Ali

---

## [Decision Letter · Decision Letter 1]

3 Mar 2026

Sociodemographic predictors of health-related quality of life dimensions in hypertensive patients: A cross-sectional study in a Ghanaian Municipal Hospital

PGPH-D-25-04048R1

Dear Mr. Darko,

We are pleased to inform you that your manuscript 'Sociodemographic predictors of health-related quality of life dimensions in hypertensive patients: A cross-sectional study in a Ghanaian Municipal Hospital' has been provisionally accepted for publication in PLOS Global Public Health.

Best regards,

Theingi Maung Maung, Ph.D

Academic Editor

Reviewer Comments (if any, and for reference):

Reviewer's Responses to Questions

**Comments to the Author**

Reviewer #2: All comments have been addressed

publication criteria?

Reviewer #2: Yes

3. Has the statistical analysis been performed appropriately and rigorously?

Reviewer #2: Yes

4. Have the authors made all data underlying the findings in their manuscript fully available (please refer to the Data Availability Statement at the start of the manuscript PDF file)?

Reviewer #2: Yes

5. Is the manuscript presented in an intelligible fashion and written in standard English?

Reviewer #2: Yes

Reviewer #2: (No Response)

**Do you want your identity to be public for this peer review?** For information about this choice, including consent withdrawal, please see our Privacy Policy

Reviewer #2: **Yes:** Dr Syed Irfan Ali
